# Sustainability Science: A Paradigm in Crisis?

**Iván González-Márquez * and Víctor M. Toledo**

Ecosystem and Sustainability Research Institute (Instituto de Investigaciones en Ecosistemas y Sustentabilidad/IIES), National Autonomous University of Mexico (Universidad Nacional Autónoma de México/UNAM), Morelia 58190, Mexico; vtoledo@cieco.unam.mx

\* Correspondence: ivan_gonzalez_marquez@hotmail.com

**Abstract:** The global socio-environmental crisis poses what is arguably the greatest challenge in the history of science. It has demanded an interdisciplinary effort in which thousands of scientists from around the world have rapidly articulated what is already recognized as a consolidated research field: Sustainability science [SS]. Considering the urgency of the matter, it is necessary to evaluate the progress so far achieved. How should this evaluation be carried out? This paper addresses this question taking into consideration some insights from the philosophy of science. In particular, it applies the conceptual framework developed by Thomas Kuhn to the study of scientific paradigms. It first reviews the development of SS, demonstrating that Kuhn's model is followed step by step. The notion of problem-solving power is discussed as the main criterion for an evaluation of scientific paradigms. Next, several elements are presented suggesting that there is a general insufficiency of problem-solving power in SS. Furthermore, additional empirical data are considered early signs of a paradigm crisis. Subsequently, the way forward for SS is discussed. From Kuhn's perspective, scientific progress is not only achieved by a steady accumulation of knowledge, but also by episodes of crises that precede radical qualitative leaps in which basic premises are modified. This paper concludes that the urgently needed progress in SS requires engaging in a critical revision of the fundamental claims upon which the field was constructed.

**Keywords:** sustainability paradigm; philosophy of science; problem-solving power; policy resistance; anomalies; paradigm crisis

---

## 1. Introduction

Science faces today what is arguably the greatest challenge of all time: A global, multidimensional socio-environmental crisis. Geo-biophysical issues recently identified by the natural sciences have been added to the long-standing social, political, and economic problems already studied by social sciences [1,2]. The convergence of all these issues confronts us with scenarios that range from severe impacts on virtually all human populations to a cascading collapse of industrial civilization. Within the context of what is the sixth mass extinction in the history of life on Earth [3], it is no exaggeration to even consider the possible extinction of the human species, the last surviving species of the group constituting the *Homo* genus.

Scientists are responding to this challenge from a diversity of disciplines. Thousands of scientists throughout the world are addressing these issues, linking through what is already acknowledged as a consolidated field of research: Sustainability science [SS] [4,5]. With only two decades of existence, it is a very recent field that has nonetheless demonstrated extraordinary growth and development [4–6]. It might seem to be too soon to attempt to assess the results achieved so far, but the urgency and seriousness of the issues demand an evaluation. Much of the existing literature reviewing the development of this field (e.g., [4–8]) focuses on data obtained through methodologies pertaining to

what is today referred to as the science of science [9]. However, is this enough to assess scientific progress? How should it be evaluated? This is a question that has been widely discussed in the field of philosophy of science, but some of the valuable insights obtained there have not been sufficiently considered in these reviews. Thus, the first objective of this paper is to bridge this gap, pointing to some elements that are relevant for this discussion. In particular, we demonstrate the usefulness of the conceptual framework developed by Thomas Kuhn for the study of scientific paradigms in order to determine the stage of progress in which SS is currently positioned, as well as proposing a way forward.

### 1.1. Paradigms and Scientific Progress: Kuhn's Model

Cited more than 110,000 times, *The Structure of Scientific Revolutions* [10] is the most famous book about science written in the twentieth century. Based on a historical study regarding the way in which some of the most important advances in the history of science have been generated, Kuhn presents a very insightful model of scientific progress resulting from a cyclical succession of paradigms and crises.

In brief, a "paradigm" is a whole way of producing science which is common to the practitioners of a particular field of research based on a shared package of fundamental claims about the world. "Normal science" is the kind of research that aims to extend and refine a particular paradigm. A normal scientist does not question those fundamental claims. Nevertheless, the sustained attention to details that characterizes normal science inevitably leads to the discovery of anomalies. "Anomalies" are problems that resist solution within the framework of a particular paradigm. At some point, anomalies may become so important that they lead to a state of "crisis," forcing the community to bring some foundations back into discussion. For Kuhn, this inaugurates a different kind of research, which he called "revolutionary science". If a new paradigm can be found with increased problem-solving power, then the community will migrate and initiate a new period of normal science (see Figure 1).

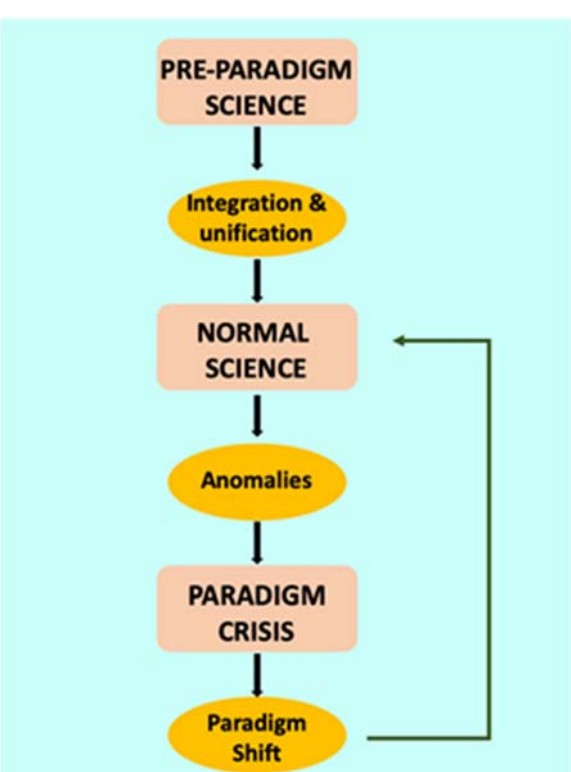

**Figure 1.** Kuhn's model of scientific progress.

Strangely, some critics consider Kuhn's work an assault on truth and progress. Some trace a direct line from him to the "post-truth" world, even blaming him for generating climate negationism. This is clearly a misunderstanding. In a later essay [11], Kuhn recalls being shocked to find out that his work was boldly misinterpreted as claiming that in science, "might makes right". As Godfrey-Smith [12] explains in an excellent introductory book, it is true that Kuhn shattered traditional empiricist ideas about how science works, but that does not mean he believed it to be a completely irrational process. Against radical relativism, he developed a subtle view about scientific change in which observation and logic occupy a very important place. He recognized some general principles used by scientists whenever they have to choose between different paradigms: They should be predictively accurate, consistent with well-established knowledge in neighboring fields, able to unify disparate phenomena, and fruitful in ideas and discoveries. In particular, he noted that problem-solving power tends to play a key role in tipping the scales in favor of one paradigm or another.

Although science is not heading toward a final "conquest of Truth," Kuhn did believe there is some real progress occurring: Our more recent paradigms have more overall problem-solving power than earlier ones, with the number and precision of solutions to problems growing over time. From this perspective, the combination of these two distinct capacities—the capacity for sustained, cooperative work and the capacity to partially break down and reconstitute itself from time to time—is essential for achieving this progress.

*1.2. Can Philosophy Help Sustainability Science (SS)?*

In the face of the planetary crisis, society trusts that science is the best tool to find solutions. However, few people would think the same about the philosophy of science. Even among scientists the prevailing opinion is that what we need today is practical knowledge to solve concrete issues rather than wasting precious time discussing general foundational issues. Addressing this exact criticism more than two thousand years ago, Aristotle fenced an argument that is particularly relevant today: The sciences in which perplexities are greater have a more extensive need for philosophy (cited in [13]). In this paper, we demonstrate that Aristotle's argument is applicable to SS.

First, we demonstrate that the development of SS up to its present state literally follows what Kuhn's model projects, demonstrating this framework's applicability and relevance to the case at hand. Next, we consider different elements that are germane to evaluation of progress in the field, arguing that, ultimately, problem-solving power is the fundamental criterion. What follows is what will probably be our most controversial claim: Counter to what is propounded by other reviews (e.g., [5–8]), we note the early signs of a coming paradigm crisis.

This controversial claim is based on the proliferation of problems that resist solution ("anomalies" in Kuhn's framework), indicating what appears to be a general insufficiency of problem-solving power. The design of solutions based on this paradigm have been inadequate to solve the aforementioned problems, which continue to worsen at an unconscionable rate. This perception is reinforced by empirical data that could be interpreted as demonstrating that scientists are already losing trust in the sustainability paradigm.

If this is the case, what is the way forward for SS? We explain exactly what it means to claim that the sustainability paradigm is in crisis and what should be done. This does not imply that the whole endeavor has failed or that we should abandon our efforts. On the contrary, this points to a need for a special kind of scientific work, different from normal science, in order to overcome the crisis and enhance SS's problem-solving power. In accordance with Kuhn's model, this cannot be achieved through the same kind of progress made in normal science. If a paradigm falls into crisis, its fundamental claims must be debated again. This kind of work needs to be performed rigorously and meticulously, demanding the participation of scholars from disciplines that have not yet been sufficiently incorporated into SS.

SS itself is facing an enormous challenge. It currently requires greater support in order to solve the urgent problems that propelled its emergence. Paradigm crises are one of the most perplexing stages in scientific progress. We are convinced that the philosophy of science can contribute important elements for guidance in these times.

## 2. Review: Sustainability as a Scientific Paradigm

A series of papers have analyzed the emergence and evolution of this field ([14–18], among others). In particular, computer-assisted *big data* methods of analysis currently make it possible to study scientific work in what is already recognized as a field of study in itself—the science of science [9].

Bibliometric analyses like those presented by Kajikawa et al. [6] offer us an initial glance into the emergence of SS. These analyses demonstrate that the number of published scientific papers that include the words sustainability or sustainable in their bibliographic records started to grow around 1990 and sharply increased after 2005, reaching 12,000 papers published annually by 2014. What we observe here is not only increasing attention from the scientific community to socio-environmental issues, but also increasing acceptance of the concept of sustainability as an appropriate way of positing the problem, pointing towards specific solutions.

Kuhn stated that paradigms emerge after an outstanding scientific work appears, which is taken to provide insights into the workings of some aspect of the world, providing a model for further investigation. Although discussions about sustainability can be found since ancient times, the way in which the concept is currently used was shaped under the notion of sustainable development [SD]. This concept appeared for the first time with the publication of the *World Conservation Strategy: Living Resource Conservation for Sustainable Development* [19]. As Kuhn stated regarding paradigms, SD is not merely a concept but a package of claims about the world. To fully appreciate the contents of this "package of claims," we need to review not only the scientific roots of SD [20], but also the technocratic and neoliberal free trade ideology inherited from development economics [21].

Among a number of alternative paradigms available at the time [22], SD became positioned in the center of international debate when it was taken up by the United Nations in its influential report *Our Common Future*, better known as the "Brundtland Report" [23]. Reviewing the evolution of SS, Bettencourt and Kaur [5] demonstrate how the cumulative number of authors in the field started growing exponentially in the years following the Brundtland Report. Using an epidemiological model, they analyzed the way in which a growing number of susceptible individuals were exposed to the SD paradigm, became "infected" and "infectious", giving way to the observed exponential growth pattern. In 1992, with the United Nations Conference on Environment and Development held in Rio de Janeiro, SD was firmly established as the official paradigm to address the global socio-environmental crisis, pervading the agendas of governments and corporations, as well as the mission of educational and research programs worldwide.

In 1999, the National Research Council called for the advent of a novel scientific discipline and coined the term "sustainability science", introducing it as "the science of sustainable development" [24]. This was followed by the Friibergh Workshop on SS in the late 2000s, leading to the publication of the seminal paper "Sustainability Science" by Kates et al. [14]. A surge of scientific journals and international conferences subsequently emerged, until the endorsement in 2017 of the United Nations 2030 Agenda for SD, which established 17 Global Goals to be achieved by 2030.

### 2.1. Pre-Paradigm Science: The Hybrid Disciplines

According to Kuhn, each scientific field starts out as a "pre-paradigm science." A paradigm is a package of claims about the world, methods for gathering and analyzing data, and habits of scientific thought and action. It is a whole way of producing science, shared by scientists working in a particular field. In the absence of this shared foundation of ideas and practices during the pre-paradigm state, scientific work is not well organized and usually not very effective. Inquiring into the emergence of a paradigm entails tracking the process through which this common foundation is constructed.

In SS, its origins were linked to what is perhaps science's greatest epistemological challenge: To overcome the *great rift* that separates the natural and social sciences. During the nineteenth century, innumerable efforts were made to generate a comprehensive understanding of natural and social phenomena, which was favored at the time by a small number of scientists. This legitimate quest—that reached its peak moment in the exchange of knowledge between the great figures of nineteenth century science (Darwin, Marx, Spencer, Humboldt, and so on)—started to dissolve by the early twentieth century, as the number of researchers grew significantly and specialization also increased. Consolidation of the largest fields of knowledge (each with its own paradigms) caused researchers, institutions, and academies to become isolated. Given that reality itself is not fragmented or segmented into disciplines, overcoming this *great rift* implies adopting a holistic, interdisciplinary, or integrative approach, as demanded for several decades by complexity science.

Today, cross-fertilization between different scientific fields has been increasingly recognized as a valuable source of new developments and innovative thinking. For this reason, inter- or multidisciplinary approaches have become more popular. Particularly towards the second half of the twentieth century, there were sometimes successful, sometimes failed attempts that went against the current of specialization and isolation of the various scientific fields. These attempts sought to connect certain areas of social sciences with the science that synthesizes the physical, chemical, geological, and biological knowledge of nature: *Ecology* or ecosystem science [25,26]. In a first stage, these attempts were addressed to the creation of a meta-science entitled *human ecology* [27,28]. This effort, however, failed to prosper and human ecology ended being considered a failed field of study [29], or marginal to SS (see [30,31]). Instead, independently, hybrid fields of knowledge emerged in at least ten areas of the social and applied sciences (Figure 2).

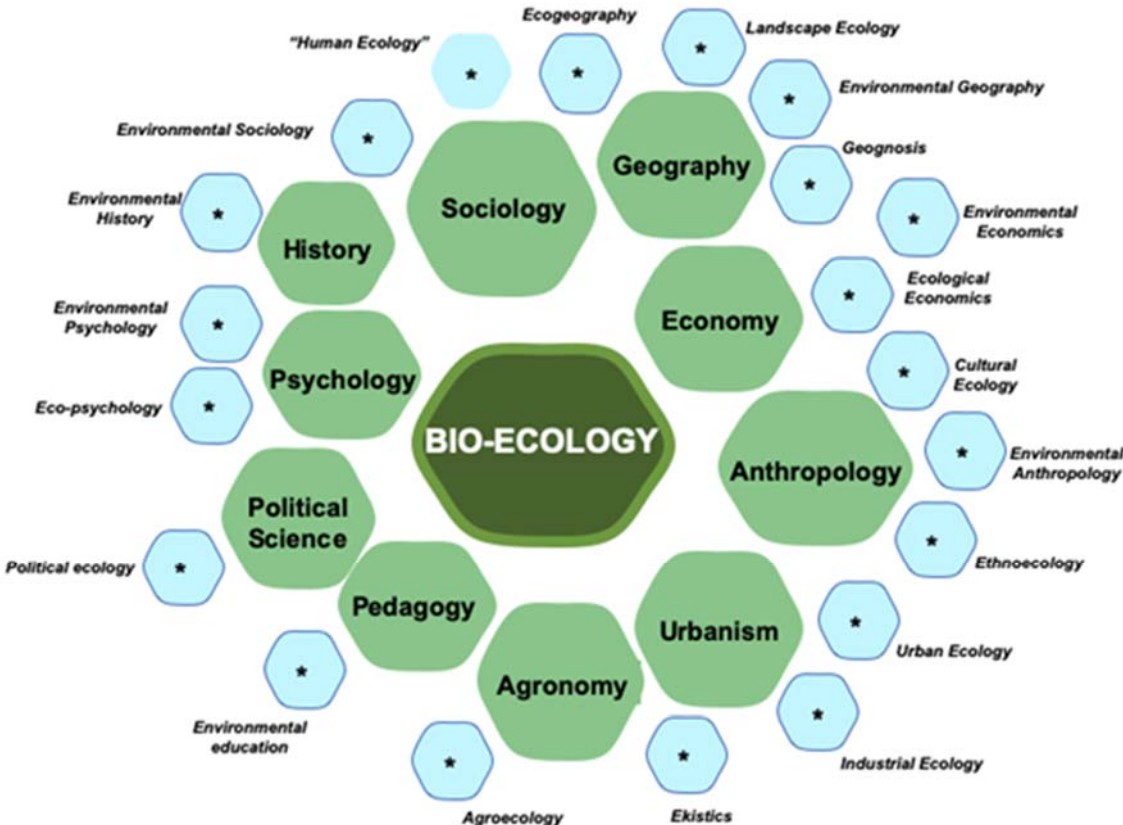

**Figure 2.** Hybrid disciplines. Modified from [32].

These *hybrid disciplines,* emerging in practically all the fields of social or applied sciences (agronomy, education, and urbanism), consider that the study of each human dimension (the social, economic, political, historical, and cultural dimensions, among others) cannot be separated from the study of nature [32]. Each has its background, history, key authors, publications, scientific societies, and academic communities that in a short time reached extraordinary levels of scientific development. Such is the case of agroecology, ecological economy, political ecology, environmental education, and environmental history.

*Hybrid disciplines* have currently adopted sustainability as their ultimate or highest goal and must therefore be considered SS precursors. Coinciding with what Kuhn notes regarding pre-paradigm science, this process has not been a coordinated, planned, or homogeneous endeavor, but rather a spontaneous and irregular convergence of a diversity of often unrelated experiences.

## 2.2. Normal Science: A Consolidated Field

Is there evidence of the formation of a shared basis of scientific practices and ideas, a whole way of doing science specific to SS beyond bibliometric analyses that show an increasing use of the concept of "sustainability" by researchers pertaining to a diversity of disciplines?

As Fang et al. [4] recall, scholars have identified various distinctive characteristics that define the particular way in which science is produced in SS, with its unique knowledge and discipline foundations, research topics, methods, and approaches. The metadisciplinary nature of the SS knowledge structure has been illustrated [15]; the research cores and topics have been bibliometrically identified [33,34]; the collaborative networks in SS have been mapped [34,35]; the knowledge base and contributing disciplines of SS have been unveiled [36]; the methodological aspect of SS has been reviewed [37–39]; and the skills and key competencies needed in SS have also been elaborated [40–43].

What kind of a science is SS? Addressing this question, Kates [44] stated that "[SS] is a different kind of science that is primarily use-inspired, as are agricultural and health sciences, with significant fundamental and applied knowledge components, and commitment to moving such knowledge into societal action". Fang et al. [4] presented an evidence-based reexamination of the same issue. Analyzing 43 definitions of SS, they identified two key elements: Understanding human–environment interactions and linking knowledge to action. These authors concluded that SS is a use-inspired, basic science of SD, which focuses on understanding human–environment interactions and linking understanding to actions by promoting a site-specific, multi-scale, and transdisciplinary approach.

Out of the heterogeneous collection of thousands of papers referring to sustainability, Kajikawa et al. [6–8] reviewed the citation networks with the purpose of evaluating the field's level of integration over time—an analysis that would be impossible for a field with such a large and rapidly developing body of literature if not for computer-assisted methodology. With this methodology, they identified the most important research domains of SS and tracked their evolution. Fewer clusters dominate the latest studies, which means that the clusters of papers that were previously separated are now more closely connected. This reveals that researchers pertaining to separate scientific communities are indeed rapidly creating a common language and understanding.

The integration of knowledge is not yet complete. Analyzing bibliometric data, Schoolman et al. [45] researched the extent to which SS lives up to its ideal of transdisciplinarity. Their results indicate that, while more interdisciplinary than other scientific fields, SS falls short of expectations in terms of the integration of environmental, economic, and social sciences. Some studies have found significant institutional and organizational impediments to interdisciplinary research [46,47] and that knowledge transfer occurs by small steps, even between neighboring subfields [48]. It is thus understandable that the *great rift* between social and natural sciences continues to represent a stumbling block that is difficult to overcome [49].

Nevertheless, there has been undeniable progress. Bettencourt and Kaur [5] sought direct evidence that SS has created a new community of practice and a new synthesis in terms of concepts and methods. Their results indicate that SS coalesced after the year 2000 in large-scale collaboration networks to which most authors now belong, producing a new conceptual and technical unification. The aforementioned elements make the case that SS has rapidly evolved into a stage characterized by many of the qualities of what Kuhn understood as normal science.

Following the work of Funtowicz and Ravetz [50], it is now common to see SS described as "post-normal science". Funtowicz and Ravetz argue that the uncertainties and high decision stakes involved in most socio-environmental problems required the emergence of a "new type of science" characterized by an extended peer community and other special qualities. Although their understanding of science's current predicament is insightful and their argumentation is compelling, their use of Kuhnian terminology is inaccurate. They misunderstand "normal science" as something that must be overcome. Strictly speaking, if Funtowicz and Ravetz were to succeed in convincing the SS community to adopt their philosophy and methodology, the paradigm of "post-normal science" would enter into a normal science stage.

As mentioned above, the capacity for sustained cooperative work under the umbrella of a particular paradigm is one of science's two essential capacities. Kuhn places great emphasis on the consensus-forging role of paradigms: Detailed work and revealing discoveries require cooperation and consensus. He claims that it is precisely because scientists doing normal science agree on a shared basis of fundamental beliefs, methods, and scientific practices that they do not waste time arguing about the most basic issues arising in their field. Against what Popper and traditional empiricists believed, he argues that there is no chance for scientists to achieve a detailed and deep understanding of phenomena nor to solve real problems unless the debate over fundamentals is eventually suspended (a key feature of normal science). With an understandable sense of urgency to solve these issues, the scientific community has sought to move towards the application of SD aimed at generating practical solutions.

### 2.3. How to Evaluate the Sustainability Paradigm?

Despite its rapid development, doubts remain regarding the potential SS has to fulfill its great ambition of fostering sustainability transitions [4]. Some authors have claimed that rhetoric outweighs real-world transitions, or even that beyond the best of intentions, the field must face the reality of failure [51]. Undoubtedly, the concept's widespread acceptance, the enormous number of published papers, and progress in unifying the field are achievements worth celebrating, but they are insufficient in claiming that SS has achieved its goal. What criteria should be used to evaluate the success or failure of a scientific field? This is a core issue widely discussed in philosophy of science. As aforementioned, beyond the specific criteria that vary from one discipline to another, Kuhn noted the existence of general principles, including problem-solving power, which is especially relevant to a problem-oriented field like SS.

It is difficult to have a direct measure of overall problem-solving power, but there are disturbing concerns. Allegedly, "silence roared" when Robert Kates asked a room of prominent sustainability scientists: "What sustainability problems have we solved over the last decade?" [51]. Kates himself recognized that there are few examples of solutions that have been offered:

> The growing body of research may at best be only slowly yielding solutions for important global and local problems in the priority action areas of population, settlements, agriculture, energy and materials, and living resources identified in the NAS report. ( . . . ) sustainability scientists are better at research than in finding and implementing solutions to local and regional sustainability problems. ( . . . ) Academic researchers also have difficulty addressing big problems in their local expression and in identifying and finding solutions. Most begin with what they already study and have difficulty to move beyond what they know to address bigger questions or to integrate their work with others [52].

Evidently, to evaluate the problem-solving power of the sustainability paradigm we must address not only the discussion about how to evaluate a research field, but also step into the realm of policy evaluation. As a practice, policy evaluation was initially rather unsystematic and mostly ad hoc. In recent decades, policy evaluation has also experienced an ongoing process of institutionalization [53,54]. To reduce the degree of subjectivity in the policy analysis process, evaluation requires systematic methodologies based on a set of normative criteria. A recent review of the literature [55] calls attention to the fact that researchers have proposed different methodologies and different evaluation criteria [56–65], however, reaching consensus about these methodologies and criteria is yet another challenge [66].

Shahab et al. [55] indicate that most outcome evaluation studies resort to the conformance-based approach (e.g., [57,62,67–71]). This approach defines a policy instrument's success or failure by means of using the degree of conformity between a policy's specified goals and its outcome. While this criterion is necessary to evaluate, it is hardly ever sufficient because it fails to study all the impacts a policy instrument might have. This conventional approach has been criticized for providing a rather biased and tunnel-vision image of the actual impact of an implemented policy instrument [55]. "Were the evaluators to confine themselves exclusively to researching the achievement of premeditated intervention goals, any serendipitous results or unanticipated side-effects would not be included in the main evaluation process" [72]. Furthermore, in an uncertain world and without all the desirable information, it is unlikely that policy objectives will be established in a totally accurate, precise, and relevant manner. Considering the dynamic and ever-changing nature of the policy environment, defined policy objectives can become irrelevant at later stages. Therefore, it is necessary to consider other evaluation models beyond the conformance-based approach [55].

The "side-effects evaluation" model [72] initially divides the effects of the policies under evaluation into anticipated and unanticipated effects occurring inside or outside the target area, and concludes with a qualitative categorization of effects. As Mickwitz [53] observes, unanticipated effects are often only partially known before an evaluation is actually undertaken. Therefore, sometimes the most important task of an evaluation is to uncover previously unknown unanticipated effects. Policies rarely turn out exactly as intended, partly owing to poor implementation, but it may also be because the drafting of policies was based on incorrect assumptions [73]. Furthermore, "policies, especially important ones, tend to be used in complex and changing contexts, where there are many other actors as well as external factors and the interactions between these are uncertain. The effects of policies, even when extensively planned, are therefore often unanticipated" [53].

In the case of sustainability policies, the kind of uncertainties acknowledged by (so-called) post-normal science challenges not only policy planning, but also policy evaluation. Indeed, social and economic systems causing the problems these policies aim to address are extremely complex and not fully understood. Therefore, although greater research does increase knowledge, environmental policies are invariably formulated and implemented in contexts within which uncertainty and lack of knowledge prevail. As concerns evaluation, this complexity implies that it is also extremely difficult to determine the effects any individual policy may have when the same system is under the influence of multiple other factors, some of which are other policies. Therefore, the whole process should be reflective and based on continuous learning from past experiences. This implies that, while it is necessary to design policies with rigorous pre-assessment preceding implementation, they should also be evaluated retrospectively more than is currently the case [53].

A final challenge is posited by the massive scale of both the socio-environmental systems and the global efforts undertaken by sustainability policies. A global evaluation demands a coordinated effort of hundreds of scientists, peer reviewers, and collaborating institutions and partners. In 1997, the UN Environment Program launched the first Global Environment Outlook (GEO). GEO reports build on sound scientific knowledge to provide governments, local authorities, businesses, and individual citizens with necessary information to guide societies towards a truly sustainable world. The most recent of these reports, the GEO-6 [74], builds on the findings of previous GEO reports, including six

regional assessments [75]. It outlines the current state of the environment, illustrates possible future environmental trends, and analyzes policy effectiveness.

*2.4. A General Insufficiency of Problem-Solving Power*

What conclusions are reached by GEO-6? Analyzing the effectiveness of the environmental policies, GEO-6 recognized that "the efforts and effects to date remain insufficient". Overall, the world is not on track to achieve neither the human nor the environmental dimension of the 2030 Agenda for Sustainable Development. And although they maintain that "well-designed environmental policies and appropriate technologies and products can often be implemented in tandem at limited or no cost to growth," they also recognize future projections showing that "development is either too slow to achieve the targets or even that it moves in the wrong direction." They conclude that "urgent action is now needed to reverse those trends and restore both environmental and human health" [74].

How can we explain that development is "moving in the wrong direction"? It seems that sometimes our "solutions" in fact aggravate our problems. In a paper included in the book *Sustainability Science: The Emerging Paradigm and the Urban Environment* [76], John Sterman, Director of the System Dynamics Group at the MIT, notices that our best efforts to solve problems often make them worse:

> As the world changes ever faster, thoughtful leaders increasingly recognize that we are not only failing to solve the persistent problems we face, but are in fact causing them. All too often, well-intentioned efforts to solve pressing problems create unanticipated "side effects." Our decisions provoke reactions we did not foresee. Today's solutions become tomorrow's problems [77].

Similar to what occurs with the well-known Jevon's paradox, Sterman here detected a perplexing pattern that is repeated in multiple examples. He called it "policy resistance": The tendency for an intervention to be defeated by the system's response to the intervention itself. Kates' perception of insufficient problem-solving power and the pattern identified by Sterman are significant since they coincide with what is observed as a general trend.

Considering 1992 as the year when SD was adopted as the official paradigm for international environmental policy, there is general agreement in pointing to the fact that after almost three decades of international environmental governance under this paradigm, social and environmental problems have continued to worsen.

At the global scale, virtually all of the socio-metabolic and Earth system indicators continued to grow at a rapid rate [1,74] following the post-1950 "great acceleration" trend [78], moving further away from sustainability. Extraction of materials, energy, and water; solid and liquid waste production; $CO_2$ emissions; climate change; ocean acidification; land degradation; biodiversity loss; all continue growing, together with economic development (GDP), human population, and urbanization [1]. An exception is the stabilization of Antarctic stratospheric ozone from about 1990. The regulation of chlorofluoro-carbons through the Montreal Protocol has been an effective human policy response to a global environmental problem [78].

At planetary scale, economic growth has not shown any significant decoupling from total energy and material consumption nor from $CO_2$ emissions [78]. Dematerialization has come true only in developed countries that outsourced industrial activity to developing countries with cheaper labor force and softer environmental regulation standards [79]. There are no signs of a global stabilization of societal resource use; rather, a new acceleration period since the early 2000s is observed [1].

Wealth concentration and inequality have increased sharply since 1980 [80]. Insofar as the human imprint on the Earth system scales with consumption, most of it is coming from the countries with the highest-income economies [78]. The number of people affected by environmental disasters is increasing, forcing millions of people to flee their homes each year, affecting disproportionately some of the more vulnerable populations [74].

Four out of nine planetary boundaries have been crossed (climate change, impacts in biosphere integrity, land-system change, and altered bio-chemical flows) pushing natural systems into unsafe operating spaces, threatening the biophysical preconditions upon which societies can exist, let alone flourish [81].

Considering this time period, 15,000 scientists stated that by failing to adequately limit population growth, reassess the role of an economy rooted in growth, reduce greenhouse gases, incentivize renewable energy, protect habitat, restore ecosystems, curb pollution, halt defaunation, and constrain invasive alien species, humanity is not taking the urgent steps needed to safeguard our imperiled biosphere. "We have learned much since 1992, but the advancement of urgently needed changes in environmental policy, human behavior, and global inequities is still far from sufficient" [82].

Perplexing as it may be, humanity has never before been moving faster nor further away from sustainability [83]. We have not found solutions able to change the general course. We have not even been able to curb the tendencies. In fact, the problems continue to worsen at a faster pace. Some have wondered whether sustainability is still even possible [84]. Other authors state that we must start talking about post-sustainability scenarios [85–87].

*2.5. Are There Early Signs of a Paradigm Crisis?*

What is a paradigm crisis? As aforementioned, typically "normal scientists" are committed to extend the paradigm and do not question its fundamental claims. Although they are not trying to find phenomena that lead to paradigm change, the sustained attention to details in an effort to apply the paradigm to more and more cases inevitably leads to the discovery of anomalies (problems that resist solution within the framework of a specific paradigm). All paradigms face some anomalies at any given time. Provided that anomalies are not numerous, normal science proceeds as usual and scientists consider them a challenge. However, anomalies tend to accumulate. Sometimes, a single anomaly gains particular prominence by resisting the efforts of the best scientists in the field. Eventually, anomalies reach a kind of "critical mass," revealing a real inadequacy in the paradigm. At this point, the field plunges into a state of crisis. Scientists stop trusting the paradigm and the most fundamental issues are brought back to the table for debate.

What Kuhn describes as an "accumulation of anomalies" is remarkably similar to the pattern identified by Sterman as "policy resistance," through which our policies not only fail to solve problems, but even make them worse. Acknowledging this trend of accumulation of anomalies or policy resistance implies that research under the sustainability paradigm is suffering a general insufficiency of problem-solving power. Now, how can we know whether this trend has already reached the "critical mass" point required to declare a paradigm crisis? According to Kuhn, a paradigm is in crisis not because someone claims so, but because the scientists in the field start to lose trust in the paradigm. Is there any empirical data supporting this claim? We believe there is. Fang et al. [4] presented some intriguing data in their analysis of the growth trends in the field (see Figure 3).

Their analysis reveals four different phases in SS development: (1) The incubation phase prior to 1999 with statistically negligible growth; (2) the emerging phase from 2000 to 2006 with steady growth; (3) the exponential growth phase from 2007 to 2012 with rapid expansion; and (4) the asymptotic (or steady) growth phase since 2013 with less quantitative growth. Beyond a merely quantitative analysis, how should we interpret this fourth stage? The authors claim it shows that "the field of SS is entering a critical stage," but their interpretation remains somewhat ambiguous. They state it may be "a sign of the emerging science losing its momentum or perishing." But they also claim that "this may indicate that SS is maturing," transitioning from "quantitative growth to qualitative development" [4].

Under the light of Kuhn's model, the end of the exponential growth phase looks like evidence of a decline in the recognition of sustainability as the adequate concept/framework to analyze problems and design solutions. Considering the aforementioned lack of problem-solving power, scientists might be losing trust in the sustainability paradigm. Does this mean that SS as a scientific field is "perishing"? In Kuhn's model, a crisis or even the breakdown of a paradigm does not imply that a

scientific community is ruined nor is ultimately failing. It just means that in order to improve our conceptual models and their problem-solving power, some fundamental issues must be brought back to the table for debate.

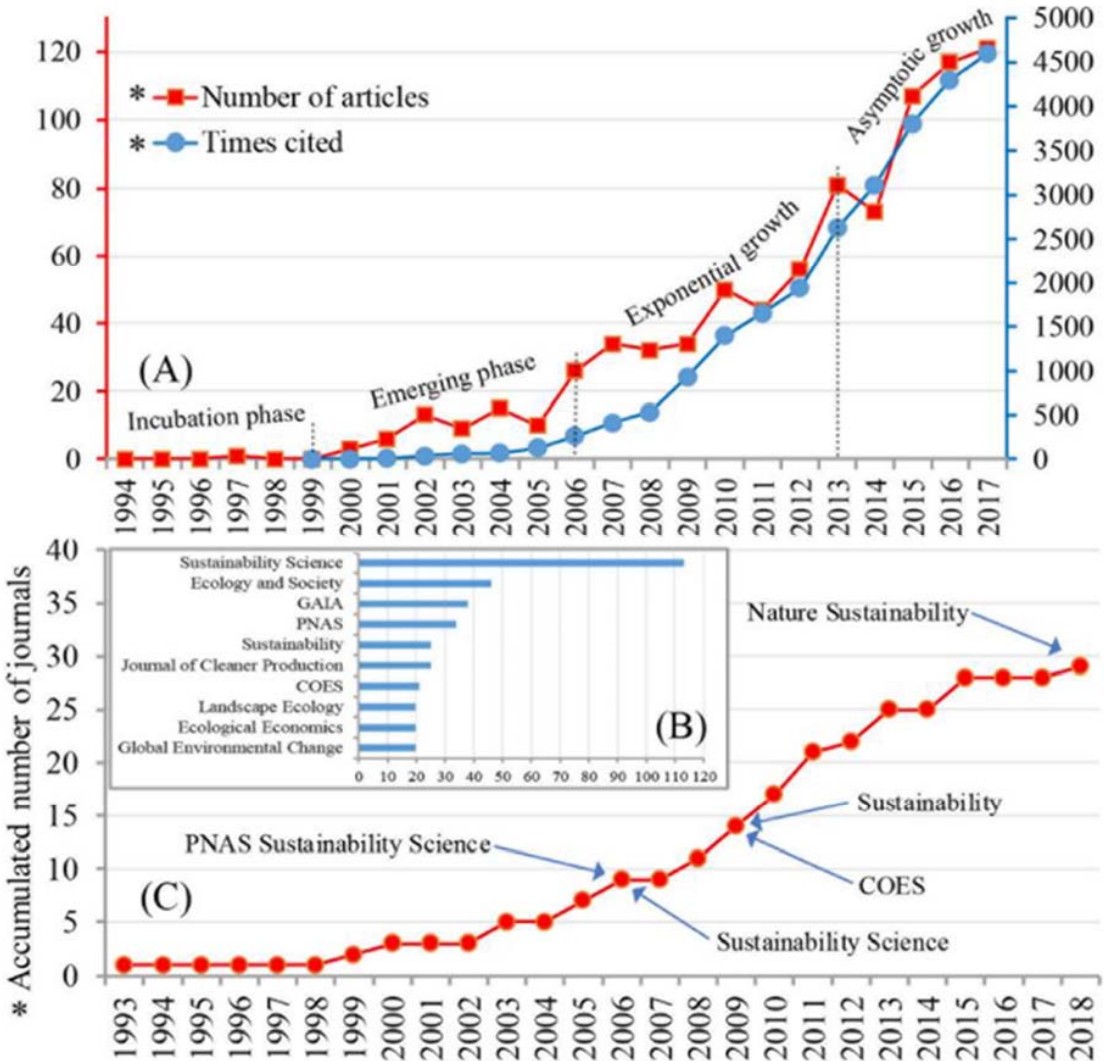

**Figure 3.** Growth trends of SS. (**A**) The number of articles on SS and their citations by year; (**B**) the top ten journals ranked by the total number of publications on SS; (**C**) the accumulative number of journals including "sustainability" in their titles. All the curves in (**A**,**C**) show an S-shaped growth pattern [4].

## 3. Discussion: What Is the Way Forward for SS?

We agree that it is urgent that science advance in order to implement solutions for the global crisis, but how should we proceed? As we have seen, there are two different ways of moving forward that depend on whether we are at the stage of normal science or facing a paradigm crisis. The elements here presented suggest that the sustainability paradigm is entering a stage of crisis.

In this situation, scientific progress requires special work. In order to identify the flaws and correct our mental models, it is necessary to explicitly identify our basic assumptions and submit them to revision. Sterman [77] reached similar conclusions: If our policies are consistently failing, we need to change our conceptual models. According to Sterman, policy resistance arises from a mismatch between the complexity of the systems in which we live and the often simplistic and erroneous mental models of the systems guiding our decisions and behavior. Talking about the "unanticipated events" and "side effects" so often invoked to explain policy failure, Sterman claims that in fact:

There are no side effects—just effects. Those we expected or that prove beneficial we call the main effects and claim credit. Those that undercut our policies and cause harm we claim to be side effects, hoping to excuse the failure of our intervention. Side effects are not a feature of reality but a sign that the boundaries of our mental models are too narrow, our time horizons too short [77].

Amusingly, Kuhn noticed that during periods of crisis perplexed scientists tend to suddenly become interested in philosophy, a field that he sees as quite useless for normal science. As Godfrey-Smith [12] explains, individual claims about the world cannot be tested in isolation. In order to test a claim, you need to make assumptions about many other aspects. So, whenever you think you are testing a single idea, what you are really testing is a long, complicated conjunction of statements and it is the whole conjunction that provides a definite prediction. If a test has an unexpected result (when an anomaly is found), then something in that conjunction is false, but the failure of the test itself does not express where the error is located. Strange observations in quantum physics, for instance, forced scientists to question very basic assumptions about the physical world, leading some researchers to think that even logic itself might need to be reviewed [88].

How does this apply to our subject matter? Under the sustainability paradigm, the kind of solutions proposed for socio-environmental problems are predominantly technological solutions or technical fixes through market-based instruments [21,77]. Technology and policies can surely be part of the solution, but if they systematically fail to solve the problems, we need to review the mental models upon which their design is based. Any specific solution that is propounded is designed within the guidelines of a certain paradigm. Any particular way of framing a problem is meaningful only within a certain worldview. If after implementing our proposed solution, we face the unexpected result of the problem continuing to persist (anomaly) or even worsening (policy resistance), then observation and logic should lead us to question our mental models.

However, this realization does not reveal where the problem lies within our mental models. What philosophers call a "holistic view of testing" implies that we should be open to critically review any part of our "web of belief" in which the problem might be found. This includes not only the level of scientific theories, but also a paradigm's fundamental claims and even the basic premises of our worldview. We must be open to question even fundamental aspects previously regarded as unquestionable: The progress of science depends on our ability to do so.

*A Call for Future Inquiry: Diving Deeply into our Web of Belief*

The predominant emphasis on technical or technological fixes implies that attention is primarily focused on discussing practical solutions and how to implement SD. The paradigm's fundamental claims are not discussed and sometimes not even acknowledged, assuming naively that they just reflect the world as it is. However, as Gómez-Bagghetun and Naredo [21] point out, "like visible parts of an iceberg," these kinds of solutions "rest on vast bodies of submerged ideology" (see also [89,90]).

Focusing on the technical level of policies, fundamental aspects of politics and economics are left undiscussed. One particular claim that has remained surprisingly unchallenged is the necessity and viability of economic growth, which is at the core of the SD consensus [21,77]. The assumption that economic growth is the universal solution to human needs and is therefore a non-negotiable necessity is a basic claim inherited from development economics. The assumption that infinite economic growth is possible rests on a deeper faith in the power of markets and human ingenuity, aptly phrased by Sterman when he claims that "modern technology's superior innovative powers and markets can compensate for any resource shortages and environmental problems growth may create" [77].

Real solutions will only be attained following an epistemological break in economic thinking. This entails a breakdown of the worldview underpinning the SD consensus, including the technological dream of dematerialization and the case for an expansionary economy premised on the axiomatic necessity of unconstrained growth [21].

We must delve even deeper into this issue. In his valuable review of the interlinking of all these elements in the "cognitive history" of the Western worldview, Jeremy Lent posited the same urgent necessity of critically studying our own mental models:

> The unconscious origin of our worldview makes it quite inflexible. That's fine when it's working for us. But suppose our worldview is causing us to act collectively in ways that could undermine humanity's future? Then it would be valuable to become more conscious of it [91].

The ideal of development taken up in SD is rooted in an older narrative of progress as the "conquest of Nature" (see [91]). In spite of its novelty as a field of research, SS carries a tacit counterproductive inheritance from the scientific revolution: As long as it keeps attempting to solve environmental problems by dealing with nature as a machine that can be governed (if we get to know its laws), it will remain inadvertently attached to the inertia that led to a global crisis in the first place.

Kates also recognizes the relevance of this theme, including it in the reader he edited for SS students [92]. After the initial overview of SD, he includes a section on human-environment interactions in which he raises the following question: "Are we part of or apart from the natural world? One set of answers, the grant by God of human dominion over nature, is derived from the Judeo-Christian tradition." At this point, he guides the students to the famous (and controversial) article, "The Historical Roots of our Ecological Crisis" [93]. In 1967, medieval historian Lynn White was already stating that this grant of dominion is what underlies the ecological crisis. However, this line of research has received insufficient follow-up in SS, which is evidenced by the fact that Kates was obliged to resort to an article published four decades earlier.

Obviously, this is a vast subject that requires much more than one article to be adequately analyzed. Here, we only insist on pointing to a shadowy area in the basement of the conceptual building of the sustainability paradigm, which needs to be brought to light. Fortunately, we do not have to start from scratch. There is a vast body of literature tracing the genealogy of the Western worldview. Exploring its vastness becomes a real challenge. That is why the effort to synthesize is so valuable. A *magnus opus* by Clarence Glacken [94], also published in 1967, is a foundational reference in the field of the history of ideas regarding the way our concept of "nature" has shaped the course of history. However, the aforementioned reference [91], pioneering the new field of cognitive history, is an up-to-date synthesis that we recommend in order to approach this discussion.

## 4. Conclusions

The socio-environmental crisis is also an epistemic crisis. Urgent scientific progress is needed. We posit that if SS resists carrying out a critical review of its own underlying assumptions, it will continue to be caught in an increasingly more powerful inertia towards collapse. In order to solve SS's problem-solving power crisis and face socio-environmental issues more effectively, it is imperative to overcome these resistances and enable "out of the box" thinking.

For SS to actually serve as "salvage science," Einstein's warning must be taken seriously; "We cannot solve our problems with the same thinking we used when we created them." However, beyond repeating the famous dictum, how can we really take this seriously? How can we address these issues rigorously? It is necessary to open a debate regarding foundational claims and for this debate we need the expertise of disciplines that have hardly been heard in the field. In the face of changing our mental framework at such deep levels, we should invite not only philosophers of science, but also specialists in worldview studies, anthropologists, and historians of ideas, among others.

We arrived at this conclusion by first demonstrating that Kuhn's model of the development of scientific paradigms is relevant and applicable to SS. The formation of this field, up to its present stage, was revealed to literally follow what Kuhn's framework stipulates. Next, we reviewed the different criteria needed to evaluate progress in a field, concluding that, ultimately, problem-solving power

is the main criterion. We looked for elements that could help evaluate the paradigm based on its problem-solving power and concluded that there is a perception of general insufficiency.

Observing what seems to be an accumulation of anomalies (problems that resist solution under the sustainability paradigm), we are sounding an alarm to warn the scientific community about the possibility of a paradigm crisis in SS. We also presented empirical, quantitative data that might be interpreted as demonstrating that the scientific community is losing trust in the paradigm. After that, we returned to Kuhn's framework to explain the exact meaning of a paradigm crisis, and what must be done to overcome it. Finally, we discussed how general guidance about how to overcome paradigm crises may apply to this particular crisis in SS, pointing to the necessity to critically review the foundational premises upon which the field is constructed. This necessary inquiry, upon which progress in the field depends at this moment of crisis, requires active participation of the aforementioned specialists.

In Kuhn's theory of science, it is only the large-scale radical changes in the way in which scientists see the world that deserve to be referred to as a "revolution," the time when one paradigm replaces another. Even then, there are different levels. Some revolutions are deeper than others, affecting greater or less extensive segments of our conceptual frameworks.

Kuhn sees the breakdown of a paradigm as part of science's "proper functioning," even though it may not feel that way to the scientists involved. The deeper and more fundamental the beliefs that are affected, the greater the resistance against a certain paradigm shift. This is due to the fact that modifying deeply held beliefs is not only intellectually and emotionally difficult, but also destabilizes knowledge and power hierarchies within a community. Nevertheless, given that this is what is needed at this point, it is imperative to overcome these resistances.

Crises and revolutions involve a breakdown, but they are essential to science as we know it. Without a crisis, scientists will not be motivated to consider radical change. Only a crisis can loosen the grip of a paradigm and make people receptive to alternatives. Revolutionary periods face a breakdown of order and a questioning of the rules of the game and are followed by a reconstruction process that can create fundamentally new conceptual structures [12].

When a paradigm enters into crisis, there is no clear map to guide the community in an orderly process towards a new paradigm. In such periods of uncertainty, scientific progress actually benefits from the unique qualities of a disorderly process, in which even very basic ideas are brought back to the table for discussion. Even epistemic standards will wobble during such chaotic moments. These are conditions that call for a special kind of creativity, the chaos from which a new direction eventually emerges.

If the seeds of a new paradigm emerge, promising to solve one or more of the problems that triggered the crisis in the old paradigm, this sudden emergence of problem-solving power sparks a revolution. A new paradigm is established when enthusiasm around a new perspective becomes the point of departure of a new tradition in scientific work followed by a new period of normal science. What happens with all the knowledge accumulated in the previous paradigm? Not everything is lost. When a revolution takes place, the most relevant and useful parts of the previous paradigm can be translated into the new one and can still be used to address the problems that they solved well (e.g., after Einstein, we continue to use Newtonian physics for innumerable practical ends).

In the face of scenarios of a probable collapse of our civilization and a possible extinction of our species, science today faces the greatest challenge of all. Perhaps the change of ideas we need is so deep that even the concept of 'paradigm shift' may fall short. Stretching concepts a little, the global socio-environmental crisis would seem to be an enormous "anomaly", a huge unexpected result of our mental model of progress, so central to the modern worldview. Will science be able to overcome this deep crisis?

We need a worldview shift, a change that reaches the ontological and cosmological foundations of our thought systems. In a subsequent paper, we will explain what this entails exactly and what are the epistemological and political implications of this shift.

**Author Contributions:** The original idea for the article, the initial literature search, and the first draft were contributed by I.G.-M., V.M.T. critically revised the first draft and conducted an additional literature search and writing. Both authors worked closely in further review and editing leading to the final manuscript. All authors have read and agreed to the published version of the manuscript.

**Funding:** This research was made possible thanks to a postdoctoral fellowship granted by the National Autonomous University of Mexico (Universidad Nacional Autónoma de México/UNAM) to Iván González-Márquez, under the guidance of Víctor M. Toledo at the Ecosystem and Sustainability Research Institute (Instituto de Investigaciones en Ecosistemas y Sustentabilidad/IIES). The publication of this article was funded by the Mexican National Science and Technology Council (Consejo Nacional de Ciencia y Tecnología/CONACyT) through the project CONACyT 5526 entitled "A National Observatory for Socio-Ecological Sustainability" ("Observatorio Nacional para la Sustentabilidad Socio-Ecológica").

**Conflicts of Interest:** The authors declare no conflict of interest. The funders had no role in the design of the study; in the collection, analyses, or interpretation of data; in the writing of the manuscript, or in the decision to publish the results.

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
