# Peer review of "Sustainability Science: A Paradigm in Crisis?"

_sustainability, doi:10.3390/su12072802_

Round 1
Reviewer 1 Report
This paper explores the concept of sustainability science using the works of Thomas Kuhn. The topic is interesting and timely. Here are my comments to improve the paper:
1- in the abstract, add a bit more discussion of your conclusions/arguments. It needs to give the reader a better understanding of what the paper is about.
2- your introduction does not sufficiently serve its purpose. You need to make the knowledge gaps clear and support your arguments with references. The introduction section needs to further engage with the literature.
3- in the introduction section, you need to define the key concepts (even briefly – although you will need to discuss them in detail later in the literature review) of your paper, e.g. paradigm, Science of Science, Sustainability Science, the theoretical framework presented by Thomas Kuhn, etc.
4- In the introduction section, make the research objectives/questions clearer. Answer the “so what?” question. Why investigating such matter is important? This is an important element that is lacking at the moment. You probably need to add 1-2 paragraphs to the introduction section to address these comments.
5- you need to further elaborate on the Kuhn’s model in section 2. Include its strengths and weaknesses (critics/limitations).
6- section 3 reads a bit descriptive. Try to be more critical/analytical in exploring these concepts.
7- table 1 is a bit odd. Why is it presented in a table format?
8- in the first paragraph of page 9 (Analyzing the effectiveness of the environmental policies…), add these references to your discussions:
Mickwitz, P. 2013. Policy Evaluation. In: Jordan, A. & Adelle, C. (eds.) Environmental Policy in the EU: Actors, Institutions and Processes. London: Routledge.
Impact-based planning evaluation: Advancing normative criteria for policy analysis. Environment and Planning B: Urban Analytics and City Science, 46(3), 534-550.
9- I am not convinced that figure 3 is necessary to include. Either remove it or elaborate further on its discussion.
10- you need to include more references in section 3.4 to support your arguments. This is an important section of your paper.
11- your conclusion section does not sufficiently serve its purpose. It is currently very short and fails to include all the key discussions and conclusions. This is a theory paper that does not use any primary data. So writing a good, longer conclusion section is critical.
Author Response
Reply to the Review Report (Reviewer 2)
- We rewrote the Abstract adding more discussion of our arguments and conclusions to give the reader a better understanding of what the paper is about.
- We added some lines in the Introduction to make the knowledge gaps clear (second paragraph of the introduction in page 1, line starting with “Much of the existing literature reviewing the development of this field [e.g. 4-8] focuses on”) and we supported the arguments with references (references 1-13).
- We rearranged the first sections (including what previously was Section 2 as subsection 1.1 of the Introduction) so that the definitions of the key concepts are included in the Introduction.
- We emphasized the research objectives and questions in the Introduction (second paragraph of the introduction in page 1, line starting with “Thus, the first objective of this paper is…”; and in the 2nd, 3rd and 4th paragraphs of the new subsection 1.2 in page 3). The new subsection 1.2 also answers the “so what” question.
- We added some lines about Kuhn’s critics on the last paragraph of page 2, (in the line that begins with: “Strangely, some critics consider Kuhn’s work…”)
- Section 3 actually is intended to be descriptive, showing that the field’s development follows step by step what would be expected according to Kuhn’s model. We added some lines in the abstract (line that says: “It first reviews the development of SS, demonstrating that Kuhn’s model is followed step by step”), in the Introduction (second paragraph of subsection 1.2 in page 3, line that says: “First, we demonstrate that the development of SS up to its present state literally follows what Kuhn’s model projects, demonstrating this framework’s applicability and relevance to the case at hand.”) and in the 3rd paragraph of the conclusions, in page 12, (lines that say: “We arrived at this conclusion by first demonstrating that Kuhn’s model of the development of scientific paradigms is relevant and applicable to SS. The formation of this field, up to its present stage, was revealed to literally follow what Kuhn’s framework stipulates.”). We hope that this makes clear what the purpose of this section is.
- We eliminated table 1 and included its content as plain text in page 8, paragraphs 5th to 9th.
- We read the two references that you suggested and we found them to be really accurate suggestions and very helpful to make subsection 2.3 much more robust. We added five whole paragraphs (starting with the 2nd in page 7) and rearranged the order of the paragraphs in the subsection to include them.
- We believe that figure 3 is very important, and we think that this subsection already explains why, but we tried to make it more clear changing the title of the subsection (“Are There Early Signs of a Paradigm Crisis?”), rephrasing the las sentences of the 3rd paragraph in page 9 (which now says: “According to Kuhn, a paradigm is in crisis not because someone claims so, but because the scientists in the field start to lose trust in the paradigm. Is there any empirical data supporting this claim? We believe there is. Fang et al. [4] presented some intriguing data in their analysis of the growth trends in the field (see Figure 3).”). The next two paragraphs (which make the whole subsection) continue discussing how we interpret the data in the graph.
- We do not have any more references to include in subsection 2.4 (in the original manuscript it was subsection 3.4), hopefully with the clarification mentioned in the previous point it will be more clear what the purpose of the subsection is: to show that the elements presented in section 2.3 already have an effect on the scientific community, evidenced in Figure 3.
- We rearranged the final sections, moving some concluding remarks from subsection 3.1 (4.1 in the original manuscript) to the Conclusions section. We also added two new paragraphs (the 2nd and 3rd in page 12) that summarize all the key discussions and conclusions.
We added more references to the discussion in section 3.1 (two new paragraphs at the bottom of page 11). We also requested a second read-through from our style correctors.
We really want to thank you for your thoughtful comments and for the extra references that you suggested, they definitely improved our paper, helping to make its objectives more clear and easy to understand.

Reviewer 2 Report
The proposed research «Sustainability Science: A Paradigm in Crisis? » falls within the scope of Sustainability. According to the reviewer’s opinion, major revisions are required in order to accept this research study for publication in Sustainability. Please, comply with the following suggestions and comments:
Comment 1: In my opinion, the aforementioned manuscript needs more data in order to be published in Sustainability. I am not so sure if your readers will find innovative data in this work.
Comment 2: The question that it needs to be answered is how it extends the existing knowledge on the topic.
Comment 3: When you submit the corrected version, please do check thoroughly, in order to avoid grammar flaws.
Author Response
Reply to the Review Report (Reviewer 2)
We made major revisions to the manuscript according to the suggestions of the reviewers, especially in the abstract, introduction, subsections 2.3, 3.1, and in the conclusions.
- This is a theory paper that does not use any primary data. From your first comment, suggesting that our paper needs more data, we conclude that we failed to make clear what are the main objectives and contributions of the paper. We hope that the changes made in the abstract, introduction (last sentences of the 2nd paragraph in pages 1-2 and especially in the new subsection 1.2, located in page 3) and conclusions (especially the newly added 3rd and 4th paragraphs in page 12), will fill that gap.
- We rewrote the Abstract adding more discussion of our arguments and conclusions to give the reader a better understanding of what the paper is about. We added some lines in the Introduction to make the knowledge gaps clear (second paragraph of the introduction in page 1, line starting with “Much of the existing literature reviewing the development of this field [e.g. 4-8] focuses on”). We emphasized the research objectives and questions in the Introduction (second paragraph of the introduction in page 1, line starting with “Thus, the first objective of this paper is…”; and in the 2nd, 3rd and 4th paragraphs of the new subsection 1.2 in page 3). The new subsection 1.2 also answers the “so what” question.
- We requested a second read-through from our style correctors.
We hope that this changes will suffice to accept our research study for publication. Thank you.

Reviewer 3 Report
Sustainability Manuscript #729680
Sustainability Science: A Paradigm in Crisis
This manuscript uses Kuhn’s theoretical model of Scientific Progress to evaluate the state of Sustainability Science. This is one of the most intellectually satisfying manuscripts I have ever had the pleasure to review. I find the manuscript exciting. The authors provide a thorough explanation of Kuhn’s theory. I appreciate the historical background on Sustainability Science. I also appreciate the discussion on the importance of interdisciplinary work and the idea of “hybrid disciplines.” The authors make an excellent argument that paradigms should be evaluated by their “problem-solving power.” The authors end the manuscript with a compelling call to action.
I agree that “[t]he socio-environmental crisis is also an epistemic crisis.” Nothing would please me more than to see the fundamental claim of the necessity of economic growth challenged. Unfortunately, given my geographic location in the U.S. with its current political context of outsized corporate power, I cannot envision this happening, therefore, I anxiously await the authors’ next paper in which they explain how this might occur. Please hurry!
The manuscript is extremely well written and interesting. It is very well sourced. I only have one very tiny comment: In the first sentence of the fourth full paragraph on page 6, I believe the correct grammatical phrasing would be “who cites whom.”
Author Response
Reply to the Review Report (Reviewer 3)
We really want to thank you for your encouraging comments, we felt that you really understood what we are trying to do with this paper and felt very happy that you are interested in the second part. This was not the case with all the reviewers so we made some major revisions to make more clear what are the objectives and key discussions in the paper.
With regards to your comment, we discussed with our style correctors what would be the best way to correct the sentence about “who cites whom” and we changed as follows: “Kajikawa et al. [6-8] reviewed the citation networks with the purpose of evaluating the field’s level of integration” (1st paragraph in page 6)
We requested a second read-through from our style correctors to check some other details as well.
Thank you very much!

Round 2
Reviewer 1 Report
Thanks for taking the comments on board. The paper reads better now and makes strong arguments. I would like to recommend the paper for publication.
Reviewer 2 Report
The authors have complied with my suggestions. Therefore the paper should be accepted for publication in its current form.